# Quantized classical response from spectral winding topology

Linhu Li [1✉], Sen Mu[2], Ching Hua Lee [2✉] & Jiangbin Gong [2✉]

Topologically quantized response is one of the focal points of contemporary condensed matter physics. While it directly results in quantized response coefficients in quantum systems, there has been no notion of quantized response in classical systems thus far. This is because quantized response has always been connected to topology via linear response theory that assumes a quantum mechanical ground state. Yet, classical systems can carry arbitrarily amounts of energy in each mode, even while possessing the same number of measurable edge states as their topological winding. In this work, we discover the totally new paradigm of quantized classical response, which is based on the spectral winding number in the complex spectral plane, rather than the winding of eigenstates in momentum space. Such quantized response is classical insofar as it applies to phenomenological non-Hermitian setting, arises from fundamental mathematical properties of the Green's function, and shows up in steady-state response, without invoking a conventional linear response theory. Specifically, the ratio of the change in one quantity depicting signal amplification to the variation in one imaginary flux-like parameter is found to display fascinating plateaus, with their quantized values given by the spectral winding numbers as the topological invariants.

[1] Guangdong Provincial Key Laboratory of Quantum Metrology and Sensing & School of Physics and Astronomy, Sun Yat-Sen University (Zhuhai Campus), Zhuhai, China. [2] Department of Physics, National University of Singapore, Singapore, Republic of Singapore. ✉email: lilh56@mail.sysu.edu.cn; phylch@nus.edu.sg; phygj@nus.edu.sg

Topological quantization has captivated a generation of physicists since the discovery of the quantum-Hall effect[1]. In the more recent years, its standing as a novel quantum phenomenon was further strengthened by the observation of discretely varying Hall conductances in the quantum anomalous Hall[2,3] and quantum spin Hall effects[4]. Rigorously established through linear response theory, the link between topological winding numbers and quantized conductivity has indeed earned a place as a classic result of quantum condensed matter physics. Indeed quantized Hall response and nontrivial Chern topology are now widely regarded to be almost synonymous.

Yet, the concept of quantized response has so far eluded classical systems. Without a quantum mechanical ground state, classical systems are not amendable to conventional linear response theory, which expresses quantized responses as perturbations upon a ground state. While classical metamaterials like photonic crystals, acoustic structures, and electrical circuits can possess an integer number of topologically protected boundary states[5–8], their response behaviors are not based on the number of accessible channels, but on analog solutions of differential equations that are by no means quantized.

In this work, we introduce the paradigm of quantized response based on the winding of the spectrum in the complex energy plane, rather than the winding of eigenstates in momentum space. For a long time, the complex spectral winding number has been exploited in predicting the non-Hermitian pumping under the open boundary conditions (OBCs) [known as the non-Hermitian skin effect (NHSE)][9–12], leading to the breaking of bulk-boundary correspondence and various anomalous topological phenomena[9–37]. In a recent experiment, arbitrary spectral winding is observed by visualizing the frequency band structure of optical frequency modes[38]. However, no directly measurable quantity has been associated with the spectral winding. Notably, spectral winding as a topological feature is not directly related to quantum physics and is hence a classical concept. Furthermore, the notion of classical response can be also a property of the Green's function alone, which has been recently shown to exhibit exponential growth in the presence of nontrivial spectral winding numbers[39–41]. While nontrivial complex energy winding is in principle well defined for quantum systems, it is most physically relevant in classical settings like mechanical, photonic, plasmonic, and electrical systems where non-Hermiticity does not present significant measurement difficulties[30,31,42]. Indeed, electrical circuits are governed by circuit Laplacians whose complex eigenvalues merely indicate phase shifts or steady-state impedances, rather than ephemeral excitations.

Specifically, what we find quantized is the response of the logarithm of the Green's function components with respect to an imaginary flux-like local parameter that continuously adjust the system boundary conditions, forming some quantum-Hall-like plateaus quantized according to the spectral winding number. This discovery was inspired by the observation that deforming a non-Hermitian system from periodic to open boundaries (PBCs to OBCs)[12,13,24,28,32], which is closely related to complex flux insertion, always reduces the spectral winding number one by one till it reaches zero. In a classical setup subject to a steady-state drive e.g., a circuit lattice with an input current, the quantized quantity can be the logarithmic impedance experienced by the response field, e.g., the voltage. This intriguing result is rooted in the way non-Bloch eigenstates explore the interior of spectral loops, and has been investigated in the new context of scale-free non-Hermitian pumping[43].

## Results

**Motivation for quantized Green's function response.** In conventional literature, quantized response usually refers to quantized

linear response in a quantum setting, where an occupied quantum state is driven by a time-dependent perturbation (see Supplemental Note 1 for a brief introduction). However, in the classical settings that we shall focus on i.e., photonic waveguides, acoustic lattices, and electronic circuit, the system does not settle into a ground state, and the Kubo formula describing the linear response, which concerns the response due to occupied quantum states, is inapplicable. While an integer number of topological modes have been observed in various photonic[7,44–46] and acoustic[47–51] settings, each classical topological mode can be excited, however, weakly or strongly depending on the energy put into it, and is thus not quantized in this sense. The classical response corresponds to the flow of arbitrary amounts of optical, phononic or electric current, and not the modified (quantized) occupancy of quantum mechanical eigenstates. Consider an external coherent drive $\boldsymbol{\epsilon} = (\epsilon_1, \epsilon_2, \ldots \epsilon_L)$ with different amplitudes applied to each of the $L$ sites of a lattice[40]. For a steady-state drive with a fixed frequency $\omega$, $\boldsymbol{\epsilon}(t) = \boldsymbol{\epsilon}(\omega) \exp(-i\omega t)$, and the resultant classical response field at the same frequency can be written as $\boldsymbol{\phi}(t) = \boldsymbol{\phi}(\omega) \exp(-i\omega t)$ with the response field amplitude given by

$$\boldsymbol{\phi}(\omega) = G(\omega, \gamma)\boldsymbol{\epsilon}(\omega), \; G(\omega, \gamma) = \frac{1}{\omega + i\gamma - H}, \quad (1)$$

analogous to the Kubo formula, which is exclusively for quantum settings. Here $G$ is the Green's function matrix and $\gamma$ represents an overall gain/loss in the system. For a signal entering the system from a single site $x$, $\boldsymbol{\epsilon}$ only possesses one nonzero component $\epsilon_x$, and the induced field at another site $y$ is $\phi_y = G_{yx}\epsilon_x$. In particular, the directional signal amplification of a signal entering one end of a 1D chain and measured at the other end is described by the two matrix elements $G_{1N}$ and $G_{N1}$[39,40].

In the spectral representation, the Green's function Eq. (1) takes the form

$$G = \sum_n \frac{1}{\omega + i\gamma - E_n} \left| \Psi_n^R \right\rangle \left\langle \Psi_n^L \right|, \quad (2)$$

where $\left| \Psi_n^{L/R} \right\rangle$ are the left/right eigenstates corresponding to the $n$th eigenenergy $E_n$. Its matrix elements $G_{xy}$ can be computed by evaluating $\left| \Psi_n^{L/R} \right\rangle$ at $x$ and $y$. Equation (2) is valid in both classical and quantum settings, since the Hamiltonian is just the operator that describes time evolution, and is well defined regardless of whether position and momentum commute.

Ordinarily, we do not expect the matrix elements of $G$ (or functions of them) to respond to an external influence $\beta$ in a quantized manner, since there is no reason why the derivatives of $(\omega - E_n)^{-1}$ and the eigenstates should conspire to add up to an integer. However, when translation symmetry is broken, the eigenstates can potentially become exponentially localized like $\sim e^{\kappa x}$, such that the matrix elements of $G$ are dominated by the largest spatial decay rate $\kappa = \kappa_{\max}$, with $G_{xy} \sim e^{\kappa_{\max}(y-x)}$. In such special scenarios, the response of $\ln G_{xy}$ for a fixed interval $x - y$ is wholly dependent on how $\kappa_{\max}$ varies with the external influence. In particular, if $\kappa_{\max}$ were to vary with an external parameter in a quantized manner, so will the response quantity $\ln G_{xy}$.

In this work, we discover that $\ln G_{xy}$ indeed possess such a quantized response if the external influence $\beta$ were to be an impurity parameter tuning the boundary conditions, which coincides with tuning an imaginary flux when the latter is sufficiently weak[12]. This quantized quantity is furthermore equal to the winding number of the energy spectrum in the complex energy plane. In the following sections, we shall elaborate on these rather surprising findings, and show the classical quantized response can be measured. While this quantization applies to both classical and quantum systems, we shall call it the quantized

classical response to distinguish it from the topological Hall response that exclusively exists in quantum systems.

**Point-gap topology and PBC-OBC spectral evolution.** To elucidate the role of spectral winding and motivate a suitable notion of classical response, we consider a generic one-band non-Hermitian system with $N$ lattice sites, described by the following tight-binding Hamiltonian

$$H = \sum_{x=1}^{N} \sum_{j=-r}^{l} t_j \hat{c}_x^\dagger \hat{c}_{x+j}, \tag{3}$$

with $t_j$ the hopping amplitude across $|j|$ lattice sites, $\hat{c}_x$ the annihilation operator of a (quasi-)particle at the $x$th lattice site, $\hat{c}_{x+N} = \hat{c}_x$ representing the PBC, and $r$ ($l$) the maximal range of the hopping toward right (left) direction. The associated momentum-space Hamiltonian is given by

$$H(z) = \sum_{j=-r}^{l} t_j z^j = P_{r+l}(z)/z^r, \tag{4}$$

with $k$ the quasi-momentum, $z := e^{ik}$, and $P_{r+l}(z)$ a $(r+l)$th-order polynomial of $z$. For any $t_j \neq t_{-j}^*$ (assuming $t_j = 0$ if $j > l$ or $j < -r$), the Hamiltonian becomes non-Hermitian and possesses a point-gap topology, characterized by a nonzero spectral winding number w.r.t. a reference energy $E_r$ enclosed by the PBC spectrum[10,11,52],

$$\nu(E_r) = \oint_{\mathcal{C}} \frac{dz}{2\pi} \frac{d}{dz} \arg \det[H(z) - E_r], \tag{5}$$

with $\mathcal{C}$ being the Brillouin zone (BZ), i.e., $k$ varying from 0 to $2\pi$. Simply put, $\nu(E_r)$ gives the number of times the PBC spectrum winds around $E_r$, as illustrated by a representative example in Fig. 1a, corresponding to the Hamiltonian of Eq. (3) with $r = l = 2$. As a side note, $\nu(E_r)$ also reflects the degeneracy of eigenstates at $E_r$ when the system is placed under semi-infinite boundary conditions (SIBC)[10].

Unlike the loop-like PBC spectrum depicting a nontrivial point-gap topology, the OBC spectrum must not cover any finite area in the complex plane, and so generically must take the form of curves within the PBC spectral loops[12,13], e.g., the Y-shape lines formed by the gray dots in Fig. 1a. That is, any reference energy $E_r$ inside a PBC spectral loop is enclosed $\nu(E_r)$ times by the PBC spectrum as $k$ varies from 0 to $2\pi$, but the same $E_r$ cannot be enclosed by the OBC spectrum. The important qualitative insight

is hence the following: if we continuously deform the system from PBC to OBC, the evolving spectrum gradually collapses from the PBC loop spectrum to the OBC line spectrum, and is therefore expected to pass $E_r$ for $\nu(E_r)$ times.

To further appreciate this understanding, we consider the real-space Hamiltonian of Eq. (3) with the following substitution only at the system's boundary, i.e., $t_j \to e^{-\beta} t_j$, if $x + j > N$ or $x + j < 1$. This introduces an additional scaling factor, or an impurity, to the boundary couplings of a 1D chain with $N$ unit cells. If one now continuously varies $\beta$ from 0 to infinity, a PBC-OBC spectral evolution can be examined in detail.

As shown in Fig. 1a, by considering only $t_{\pm 1}$ and $t_{\pm 2}$, we already have a representative and intriguing model whose spectral winding number can be either 1 or 2 in the topologically nontrivial regime. Next we consider the tuning of the boundary coupling via $t_{\pm 1} \to e^{-\beta} t_{\pm 1}$ for $x = N$ and $t_{\pm 2} \to e^{-\beta} t_{\pm 2}$ for $x = N, N-1$. Let the $n$th right eigenvalue of this model system with such PBC-OBC interpolations be $E_n(\beta)$. Then the spectral evolution is all captured by $E_n(\beta)$ vs $\beta$. To motivate the connection with the Green's function, and to capture incidences when $E_n(\beta)$ comes close to $E_r$, we also define a quantity

$$I_\beta(E_r) = \sum_n |1/(E_n(\beta) - E_r)|, \tag{6}$$

i.e., the absolute sum of the inverse energy spacings between the evolving spectrum $E_n(\beta)$ and a reference energy $E_r$. For an actual system always of finite size, $E_n(\beta)$ is discretized, but it can still be made to be very close to the PBC reference eigenvalue $E_r$ for a sufficiently large system. As such, the quantity $I_\beta(E_r)$ can be a diagnosis tool to examine how many times $E_n(\beta)$ visits (the proximity of) $E_r$ as $\beta$ is tuned. Moreover, the OBC limit can be essentially reached once $\beta$ is beyond a critical value $\beta = \beta_{OBC} \sim N\eta$ with $\eta$ the effective localizing length of the eigenstates[43,53]. With these understandings, one infers that as $\beta$ varies from 0 to $\beta_{OBC}$, $I_\beta(E_r)$ is expected to display high peaks whenever the complex spectral evolution passes through $E_r$. As explained above, the total number of such local peaks then reflects the spectral winding number $\nu(E_r)$. In Fig. 1b we illustrate $I_\beta(E_r)$ as a function of $\beta$ for several $E_r$ denoted by the stars of different colors in Fig. 1a, corresponding to different spectral winding numbers $\nu(E_r)$. It is indeed observed that the number of peaks of $I_\beta(E_r)$ directly reflects the spectral winding number $\nu(E_r)$. In the Methods section, we offer more insights based on the so-called generalized Brillouin zone (GBZ), to better understand why $\nu(E_r)$

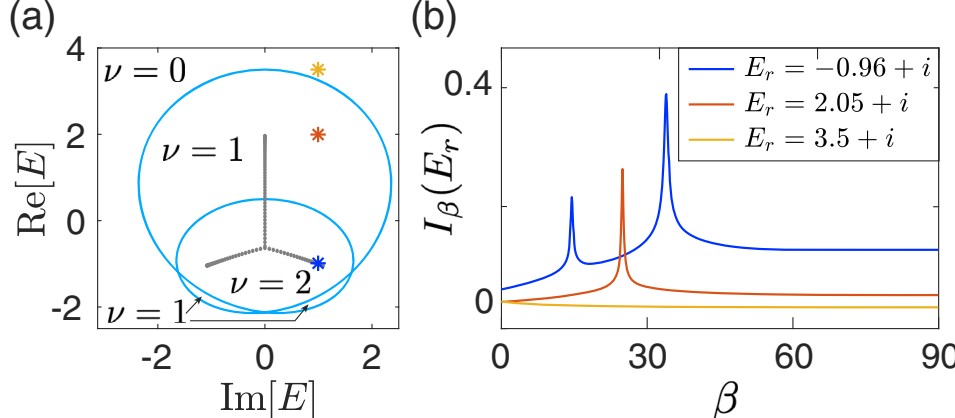

**Fig. 1 The generic Hatano-Nelson model with different spectral winding numbers. a** Spectra under periodic (cyan loops) and open (gray dots) boundary conditions, of the model [Eq. (3)] with hopping ranges up to 2. The spectral winding number for each different regime is indicated on the panel. **b** The quantity $I_\beta(E_r) = \sum_n |1/(E_n(\beta) - E_r)|$, where $E_r$ is the reference energy marked in (**a**) with the same color (yellow, red, and blue), and $E_n(\beta)$ is the $n$th eigenenergy of the system with the hoppings across the boundary multiplied by $e^{-\beta}$. $I_\beta(E_r)$ diverges exactly $\nu$ times when $E_r$ sits in a region of spectral winding $\nu$. Parameters are set at $t_1 = 1$, $t_{-1} = 0.5$, $t_2 = 2$, and $t_{-2} = 0$, with $N = 100$.

can be captured by the number of singularities encountered throughout the complex spectral evolution.

**Quantized response in signal amplification.** While the previously defined quantity $I_\beta(E_r)$ is useful in diagnosing the spectral winding, it is not directly measurable. Below, we show how it inspires another analogous quantity that is directly associated with signal amplification. In particular, the quantity introduced below displays quantized plateaus that precisely match spectral winding numbers, making it possible to distinguish between one nontrivial point-gap topology from another. This is a true advance as compared with earlier interesting attempts where signal amplification was only used to probe NHSE under OBC[39,40].

For our system with the PBC-OBC interpolation parameter $\beta$ ($0 < \beta < \beta_{\text{OBC}}$), what enters into the expression of the Green's function $G$ in Eq. (2) is $1/[E_r - E_n(\beta)]$ again (as in $I_\beta(E_r)$) and hence the Green's function should be able to capture the complex spectral evolution. More importantly, it is found that the associated eigenstates under PBC-OBC interpolation pile up at its boundary in a manner qualitatively different from that of NHSE[43]. With further derivation in the Method section, we find the amplification ratio from site $N$ to site 1

$$|G_{1N}| \propto e^\beta, \quad \frac{d\ln|G_{1N}|}{d\beta} = 1 \tag{7}$$

for a one-dimensional system with only the nearest-neighbor couplings, provided $t_1 > t_{-1}$ and $E_r$ falls within the loop spectrum of the system at a given $\beta$ (see Method). For $t_{-1} > t_1$, one can analogously obtain $\frac{d\ln|G_{N1}|}{d\beta} = 1$, corresponding to a winding number $\nu(E_r) = -1$. We emphasize that the variation in Eq. (7) can be directly measured in a steady-state response experiment. In an electrical circuit setting, this task can be done via impedance measurements, as will be elaborated later.

For systems with solely $m$th-nearest-neighbor coupling, our analysis detailed in Methods section shows that the system can be viewed as $m$ independent subchains and the overall amplification can be captured by the $m \times m$ diagonal block at the top-right (bottom-left) corners of the overall Green's function $G$, denoted as $G_{\leftarrow,m\times m}$ ($G_{\rightarrow,m\times m}$), corresponding to measuring the output at the first (last) $m$ sites of a signal entering from the last (first) $m$ sites. Since each subchain yields an amplification factor proportional to $e^\beta$, one directly obtains that $|G_{\leftarrow,m\times m}| \equiv \det[G_{\leftarrow,m\times m}] \propto e^{m\beta}$ or $|G_{\rightarrow,m\times m}| \equiv \det[G_{\rightarrow,m\times m}] \propto e^{m\beta}$, depending on the amplification direction. Specifically,

$$\nu_{\leftarrow,m} \equiv d\ln|G_{\leftarrow,m\times m}|/d\beta \tag{8}$$

or analogously $\nu_{\rightarrow,m} \equiv d\ln|G_{\rightarrow,m\times m}|/d\beta$ is quantized at $m$. Clearly then, $\nu_{\leftarrow,m}$ ($\nu_{\rightarrow,m}$) counts the number of independent amplified modes whose amplification factor has the $e^\beta$ dependence. On the other hand, the PBC spectrum of the system winds $m$ times around the origin of the complex plane, a fact obviously true since the associated momentum-space Hamiltonian is dominated by the terms $e^{\pm imk}$. For more general cases with coexisting couplings across $r$ ($l$) lattice sites to the right (left), the system can still be effectively understood as $m = \text{Max}[r, l]$ different subchains, with $t_{\pm m}$ viewed as the nearest-neighbor coupling on each subchain, and the rest understood as inter-subchain or intrachain longer-range couplings. For example, lattice sites $j$, $j+m$, $j+2m, \cdots$, with each pair of neighbors coupled by $t_{\pm m}$, form the $j$th subchain, for $j = 0, 1, 2, \cdots, m-1$. Although the subchains are now coupled, the above-defined $\det[G_{\leftarrow,m\times m}]$ ($\det[G_{\rightarrow,m\times m}]$) remains physically relevant: given by the product of all its eigenvalues, it represents the product of the

amplification factors of all the independent eigenmodes. Therefore, $\nu_{\leftarrow,m}$ ($\nu_{\leftarrow,m}$) still counts the number of effectively independent amplified modes with the amplification factor $e^\beta$. As such, there is a fascinating correspondence between quantized response as captured by $\nu_{\leftarrow,m}$ or $\nu_{\rightarrow,m}$ and the spectral winding number.

Figure 2 presents computational results for $\ln|G_{\leftarrow,2\times 2}|$ and its derivative with respect to $\beta$ as functions of $\beta$, denoted $\nu_{\leftarrow,2}$, again for the same system as that in Fig. 1. (A more complicated example with hopping range up to 3 is demonstrated in Supplemental Note 2.) We also compare these results with $I(E_r)$ defined previously for a chosen reference energy point $E_r = \omega + i\gamma$, with $\nu(E_r) = 2$. It is observed that $\nu_{\leftarrow,2}$ as a measurable physical response shows three clear plateaus quantized at $\nu_{\leftarrow,2} = 2, 1, 0$. Echoing with the jumps between these plateaus during the spectral evolution, $I_\beta(E_r)$ shows local peaks whenever the spectral evolution passes through $E_r$. As shown in Fig. 2c, these transitions during the spectral evolution as a result of increasing $\beta$ match precisely with the $\beta$ values for which the complex spectrum touches the reference energy point and hence the spectral winding number is about to jump. That is, the transitions between these quantized plateaus have a clear topological origin and are hence identified as topological transitions. As a side remark, observing all the plateaus in our example here happens to require a broad regime of $\beta$ and hence cases with very weak boundary coupling. This is, however, not a concern because the first plateau at small $\beta$ is the one to reflect the topology under PBC. Being a topologically quantized response, these plateaus are also robust against disorder, as shown in Supplemental Note 3.

The results presented in Fig. 2 are particularly stimulating. Indeed, therein neither the next-nearest-neighbor coupling nor the nearest-neighbor coupling is dominating. Yet quantized plateaus at $m = 1$ and $m = 2$ are still obtained. Returning to our decoupled subchain picture above, this indicates that for different reference energy points, the behavior of this system is topologically equivalent to that of one single chain or that of two weakly coupled subchains. With this perspective, we propose to examine $\nu_{\leftarrow,m}$ vs. different choices of $m$ in order to fully map out the phase boundaries, without any prior knowledge of spectral winding. We thus proceed to find the maximal $\nu_{\leftarrow,m}$ by scanning $m$, for sufficiently small $\beta$. The obtained value is then expected to yield $\nu(E_r)$ of the studied system under PBC. To this end we define $\nu_\leftarrow = \text{Max}[\nu_{\leftarrow,1}, \nu_{\leftarrow,2}]$, given that our model system at most has effectively two subchains (one can similarly define $\nu_\rightarrow$). We present in Fig. 3a our results of $\nu_\leftarrow$ at $\beta = 0$ for different $\omega$ and $\gamma$, with the reference energy $E_r = \omega + i\gamma$. The phase boundaries identified there are in excellent agreement with the actual PBC spectrum shown in Fig. 1a. The only subtlety is that we also obtain a negative value $\nu_\leftarrow = -1$ in the topological trivial regime. In fact, the quantity $\nu_\leftarrow$ does not contain the topological information in this regime, as $\nu_{\leftarrow,0}$ is not defined in our formalism. In retrospect, the central quantity $\nu_{\leftarrow,m}$ with arbitrarily chosen $m$ as a positive integer is expected to yield a nonpositive value in this regime, because of the absence of a directional amplification for $E_r$ with zero spectral winding (more details in the Method section). That is, in these topologically trivial regimes both $\nu_\leftarrow$ and $\nu_\rightarrow$ are found to be negative, the amplification factor is far less than unity, and hence there is actually no signal amplification after all. One may just exploit this additional feature to locate regimes with zero spectral winding. By contrast, a topologically nontrivial regime with negative winding $\nu(E_r) < 0$ corresponds to the directional amplification toward the opposite direction, and can be measured by $\nu_{\rightarrow,m}$.

Our qualitative analysis and quantitative results are so far based on single-band systems. To establish a more general and intriguing connection between spectral winding topology and

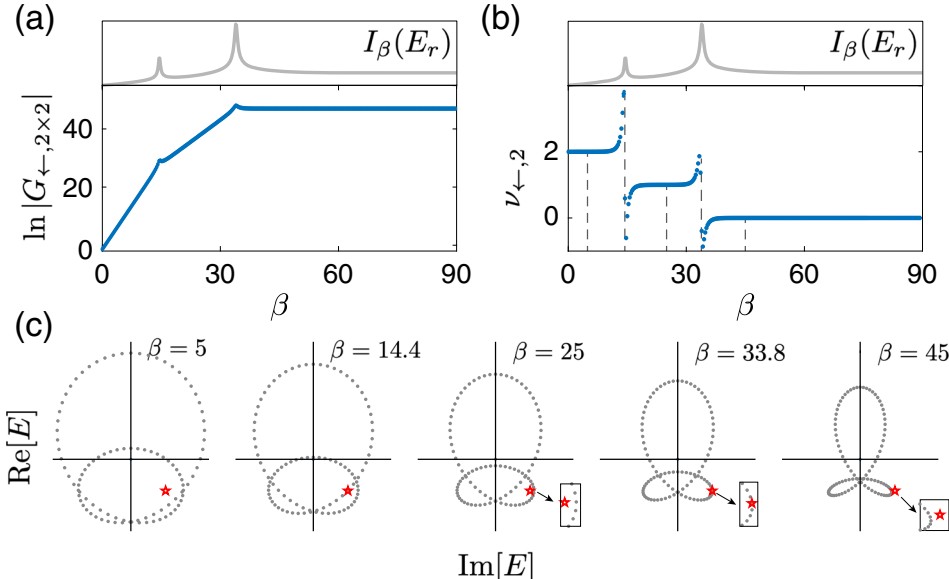

**Fig. 2 Quantized plateaus in the interpolation of the system between periodic and open boundary conditions. a** Amplification ratio $|G_{\leftarrow,2\times2}|$ given by the of the two-by-two off-diagonal block of the Green's function, and **b** $\nu_{\leftarrow,2} = \partial\ln|G_{\leftarrow,2\times2}|/\partial\beta$, as a function of $\beta$, with $\beta$ controlling the boundary condition, along with the previously defined $I_\beta(E_r)$, whose peaks match well with the jumps between different plateaus. **c** Spectrum at different values of $\beta$, corresponding to the five dashed lines in (**b**), respectively. Red stars indicate the reference energy $E_r = \omega + i\gamma$, with $\omega = -0.96$ and $\gamma = 1$. Insets zoom in on the regime around $E_r$ to give a clearer view of the relation between the shown spectrum and $E_r$. Parameters are set at $t_1 = 1$, $t_{-1} = 0.5$, $t_2 = 2$, $t_{-2} = 0$, and $N = 100$.

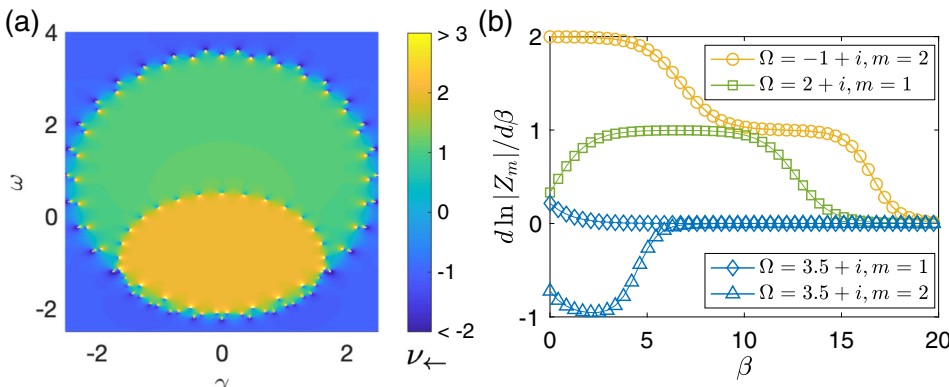

**Fig. 3 The topological phase diagram and simulation of impedance measurement in a circuit realization. a** The topological phase diagram, in excellent agreement with that shown in Fig. 1, is mapped out from examining physical response functions $\nu_{\leftarrow} = \mathrm{Max}[\nu_{\leftarrow,1}, \nu_{\leftarrow,2}]$ for the model depicted in the main text, with parameters set at $t_1 = 1$, $t_{-1} = 0.5$, $t_2 = 2$, $t_{-2} = 0$, and $N = 100$. The reference energy for defining $\nu_{\leftarrow}$ is given by $E_r = \omega + i\gamma$. At the phase boundary, $E_r$ is an eigenvalue of the system under the periodic boundary condition, leading to the divergence of the Green's function matrix. **b** Quantization of $\frac{d\ln|Z_m|}{d\beta}$, which is directly obtainable from simulated circuit measurement data of the impedance between the first $m$ and last $m$ nodes of the circuit lattice, here computed with 50 nodes. $Z_m$ (Eq. (10)) is the $m \times m$ matrix of impedances between the measured nodes, and $\beta$ is the parameter controlling the strengths of the couplings connecting the two ends. The complex admittance parameter $\Omega$ indicated on panel **b** is connected with parameters $\omega$ and $\gamma$ plotted in panel **a** with $\omega = \mathrm{Re}(\Omega)$ and $\gamma = \mathrm{Im}(\Omega)$. The first plateau encountered when $\beta$ is increased from zero gives the nonzero topological winding $\nu$ (green, yellow for $\nu = 1, 2$). When $\nu = 0$, the logarithmic gradients of both $Z_{1,2}$ are close to or smaller than 0 (blue), corresponding to the absence of signal amplification toward either direction.

signal amplification, one must verify whether quantized response still emerges in multiband systems. The answer is affirmative. This is a remarkable finding because in multiband systems, the spectral winding topology can be jointly induced by different bands on the complex plane connected all together, leaving each individual band contribute some fractional winding only. To treat multiband systems, we consider the analogous $m \times m$ Green's function block by identifying one fixed sublattice of each unit cell, a natural route because the steady-state profile on the chosen sublattice (sublattice chosen to have nonzero support of the steady state) from one unit cell to another does contain

information of amplification. As an example for demonstration, we have investigated an extended Su-Schrieffer-Heeger[54] model with a high-spectral winding number in a variety of parameter regimes, with plateaus corresponding to different spectral winding numbers. [See Supplemental Note 4].

**Measurement of quantized response in electrical circuits**. We next elaborate on how the quantized response can be directly extracted by measuring the impedance in an electrical circuit setting. Instead of external perturbations, a circuit is most

naturally driven by a steady-state AC or DC current, with voltage response given by Kirchhoff's law $\mathbf{I} = L\mathbf{V}$, where $L$ is the circuit Laplacian matrix and the components of $\mathbf{I}$ and $\mathbf{V}$ are, respectively, the input currents and voltages at each node. Suppose that the circuit is then grounded by identical circuit components with complex admittances $-\Omega$. In this case, the full (grounded) Laplacian becomes $J = L - \Omega\,\mathbb{I}$, and the voltage distribution due to the input current are given by[55,56]

$$\mathbf{V}_i = [J^{-1}]_{ij}\mathbf{I}_j = -G_{ij}\mathbf{I}_j = \left[\sum_n \frac{|\Psi_n^R\rangle\langle\Psi_n^L|}{E_n - \Omega}\right]_{ij}\mathbf{I}_j \qquad (9)$$

where the eigenvalues $E_n$ and L/R eigenstates $|\Psi_n^{L/R}\rangle$ are that of the Laplacian $L$. Notably, the quantity in the square parentheses agree exactly with our definition of the Green's function (Eq. (2)), with $\Omega$ taking the role of $\omega + i\gamma$, analogous to a classical version of a tunable "Fermi energy" for probing the physics around a reference energy. We extract the physical response through the impedance $Z_{ij}$ between the $i$th and $j$th nodes, which is related to the Green's function via $Z_{ij} = G_{ij} + G_{ji} - G_{ii} - G_{jj}$. By varying the identical grounding admittances $\Omega$ via a combination of RLC components with $\pm\pi/2$ relative phase shifts, we will be able to effectively access the response from different regions of the complex spectral plane.

Quantized classical response can be obtained from a circuit whose Laplacian exhibits nontrivial spectral winding. This requires effectively asymmetric couplings between nodes, which has already been demonstrated in existing topolectrical experiments through combinations of capacitors, inductors and INICs (negative impedance converter with current inversion) comprising operation amplifiers[30,57,58], as further elaborated in the Supplemental Note 5. The boundary couplings can also be adjusted to effect the variation with $\beta$ through tunable inductors connected in series with the asymmetric couplings[43]. Arbitrarily large spectral winding numbers can always be achieved by coupling sufficiently distant nodes, which can be much more feasibly done in electrical circuits compared to other platforms.

In analogy to $\nu_{\leftarrow,m}$ from Eq. (8), we can define $Z_m = \det Z_{ij}|_{i \leq m, N-j < m}$, the determinant of the $m \times m$ matrix of impedances between the first $m$ nodes at one end with the last $m$ at the other end of the circuit chain. Although it is not exactly equivalent to $\nu_{\leftarrow,m}$, it is expected to vary with $\beta$ in a similar manner, since the impedance $Z_{ij}$ is dominated by the component of the Green's function that produces the directional amplification. Therefore, by keeping track of the effective $\beta(\omega)$ and $\Omega(\omega)$, the logarithmic gradient of the impedance determinant $\frac{d\ln|Z_m|}{d\beta}$ will be expected to exhibit quantized jumps as $\Omega(\omega)$ crosses the boundary between regions of different topological winding $\nu$. For $m \leqslant 2$, which includes the model we had considered (Figs. 1, 2, and 3), we explicitly have

$$Z_{m=2} = Z_{1,N-1}Z_{2,N} - Z_{1,N}Z_{2,N-1}, \qquad (10)$$

whose gradients are dominated by those of terms like $G_{1,N-1}G_{2,N}$ and $G_{1,N}G_{2,N-1}$ in the presence of directional amplification (toward the first lattice site). As demonstrated via the simulated measurements in Fig. 3b, the gradient $\frac{d\ln|Z_{m=1,2}|}{d\beta}$ indeed exhibits plateaus quantized at the winding number $\nu$ (blue, green, yellow for $\nu = 0, 1, 2$, respectively) where $\Omega$ is tuned to. For higher topological winding, we take the plateau that first occurs when $\beta$ is increased from 0, i.e., the plateau closest to periodic boundary conditions.

## Discussion

In this work, we have introduced the new paradigm of quantized classical response, where a quantized response coefficient is established from a subblock of the Green's function matrix that varies with an imaginary flux-like parameter. Being based on the topological winding properties of the Green's function in the complex energy plane, this quantization does not assume the existence of any quantum mechanical ground state, and applies to all systems, classical, and quantum. Specifically, in a variety of situations including multiband cases where the spectral winding topology is rich, we show that the spectral winding number is directly detectable as a steady-state response coefficient to changes in the boundary condition. Indeed, through the signal amplification setting, the number of independent amplifiable modes that share a common exponential dependence on the imaginary flux-like parameter can be experimentally determined, which reveals the spectral winding number. Such correspondence between spectral winding numbers and quantized response is arguably broader in scope than in the case of momentum-space topology, because spectral winding does not even require translational invariance. Our results are relevant to a number of current experimental platforms of non-Hermitian systems[30,31,33,39,59–61]. In the context of classical electrical circuits, we have shown that a quantized response can be easily extracted from extremely experimentally accessible impedance measurements.

## Methods

**Insights based on generalized Brillouin zone.** Here we use the so-called generalized Brillouin zone (GBZ) method to elaborate why $\nu(E_r)$ can be captured by the complex spectral evolution. According to the non-Bloch band theory, the OBC spectrum can be described by the PBC one in a GBZ, using a complex deformation of the quasi-momentum $k \to k + i\kappa_{\mathrm{OBC}}(k)$[9,12,32,62,63]. The PBC-OBC spectral evolution can then be effectively described by $k \to k + i\kappa(k)$ with $\kappa(k)$ varying from 0 to $\kappa_{\mathrm{OBC}}(k)$, with $\kappa_{\mathrm{OBC}}(k)$ having the minimal magnitude to yield the OBC spectrum[12,63]. The PBC-OBC spectral evolution can hence be understood as arising from tuning $\kappa(k)$ and hence deforming the BZ to the GBZ, as shown in Fig. 4. Moreover, an winding number can be defined as,

$$\nu(E_r) = \oint_{\mathrm{GBZ}} \frac{dz}{2\pi} \frac{d}{dz} \arg \det[H(z) - E_r], \qquad (11)$$

analogous to that of Eq. (5), with integration in the GBZ instead of the BZ. This winding number must be zero when the OBC spectrum is reached because, again, the OBC spectrum cannot enclose any finite area[10,11]. Following the Cauchy principle, the spectral winding number is found to be $\nu(E_r) = N_{\mathrm{zero}} - N_{\mathrm{pole}}$, where $N_{\mathrm{zero}}$ and $N_{\mathrm{pole}}$ are the counting of zeros and poles enclosed by the integration path (BZ or GBZ) weighted by their respective orders. The conclusion is hence as simple as follows. If we continuously tune $\kappa(k)$, the PBC-OBC spectral evolution must pass through different zeros of $\left[\frac{P_{r+l}(z)}{z^r} - E_r\right]$ [colored dots in Fig. 4] for a total of $\nu(E_r)$ times, such that the spectral winding number reduces from $\nu(E_r)$ to 0 eventually when the integration path approaches the GBZ. Thus, during the complex spectral evolution, the spectrum under $k \to k + i\kappa(k)$ must pass the reference energy $E_r$ for a

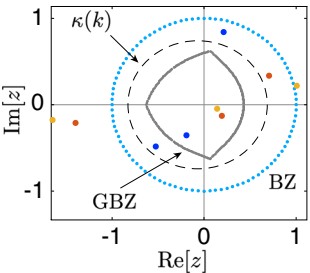

**Fig. 4 A typical example of the Brillouin zone (BZ) and the generalized Brillouin zone (GBZ).** The BZ and GBZ are given by cyan and gray dots, respectively, in the complex plane of $z := e^{ik}e^{-\kappa(k)}$. Here $\kappa(k)$ represents a complex deformation of the momentum $k$. The black dash loop indicates an evolving GBZ with $\kappa(k)$ between 0 and $\kappa_{\mathrm{OBC}}(k)$, the value that gives the spectrum under the open boundary condition. Blue, red, and yellow dots are the zeros of $H(z) - E_r$, where $H(z)$ is the PBC Hamiltonian of the system and $E_r$ is the chosen reference energy for calculating the winding number. The system is chosen as the same as that in Fig. 1 in the main text, i.e., $H(z) = 2z^2 + z + 1/2z$.

total of $\nu(E_r)$ times, constituting a rather formal argument to justify our treatment in the main text.

## Directional signal amplification versus the PBC-OBC spectral evolution

*Cases with only nearest-neighbor coupling.* As mentioned in the main text, under an external drive $\boldsymbol{\epsilon}(t) = \boldsymbol{\epsilon}(\omega) \exp(-i\omega t)$ and an overall on-site gain/loss parameter $\gamma$, the resultant response field $\boldsymbol{\phi}(t)$ can be written as $\boldsymbol{\phi}(t) = \boldsymbol{\phi}(\omega) \exp(-i\omega t)$, with

$$\boldsymbol{\phi}(\omega) = G(\omega, \gamma)\boldsymbol{\epsilon}(\omega), \quad G(\omega, \gamma) = \frac{1}{E_r - H}, \quad (12)$$

where $E_r = \omega + i\gamma$, $G$ is the Green's function matrix. The amplification factors for a signal toward the left and the right are described by the matrix elements $G_{1N}$ and $G_{N1}$, respectively. For a non-Hermitian $H$, the Green's function matrix can be expressed in the spectral representation[40,41]

$$G(\omega, \gamma) = \frac{1}{E_r - H} = \sum_n \frac{1}{E_r - E_n^R} \left| \Psi_n^R \right\rangle \left\langle \Psi_n^L \right|, \quad (13)$$

with $\left| \Psi_n^R \right\rangle$ the $n$th right eigenstate of $H$ with eigenenergy $E_n^R$, and $\left\langle \Psi_n^L \right|$ the corresponding left eigenstate.

To be more explicit, consider the Hatano-Nelson model under the PBC-OBC interpolation, described by the following Hamiltonian

$$H_\beta = \sum_{x=1}^{N-1} (t_1 \hat{c}_x^\dagger \hat{c}_{x+1} + t_{-1} \hat{c}_{x+1}^\dagger \hat{c}_x) + e^{-\beta} (t_1 \hat{c}_N^\dagger \hat{c}_1 + t_{-1} \hat{c}_1^\dagger \hat{c}_N), \quad (14)$$

also with the assumption $t_1 > t_{-1}$ without loss of generality. We shall also assume $N \gg 1$ as topological properties are usually more significant in large systems. Let the $n$th right eigenstate be $\left| \Psi_n^R \right\rangle = \sum_{x=1}^\infty \psi_{x,n}^R \hat{c}_x^\dagger |0\rangle$, with $|0\rangle$ the vacuum state. Using the eigenvalue-eigenstate equation $H_\beta \left| \Psi_n^R \right\rangle = E_n^R \left| \Psi_n^R \right\rangle$, one obtains

$$t_1 \psi_{x+1,n}^R + t_{-1} \psi_{x-1,n}^R = E_n^R \psi_{x,n}^R \quad (15)$$

for $x \in [2, N-1]$, and

$$e^{-\beta} t_1 \psi_{1,n}^R + t_{-1} \psi_{N-1,n}^R = E_n^R \psi_{N,n}^R, \quad (16)$$

$$t_1 \psi_{2,n}^R + e^{-\beta} t_{-1} \psi_{N,n}^R = E_n^R \psi_{1,n}^R. \quad (17)$$

Taking the following ansatz eigensolutions:

$$\psi_{x,n}^R = C_n e^{-M_n^R (x-1)} \quad (18)$$

with $C_n$ the normalization constant, we obtain

$$e^{-\beta} \frac{t_1}{t_{-1}} + e^{-M_n^R(N-2)} = e^{-\beta} e^{-M_n^R(2N-2)} + \frac{t_1}{t_{-1}} e^{-M_n^R N}. \quad (19)$$

Now let us discuss the possible range of $M_n^R$ depending on $\beta$, $t_1$, and $t_{-1}$. We first note that if $\mathrm{Re}[M_n^R] < 0$, then the two sides of Eq. (19) exponentially explode to infinity with different rates when $N \gg 1$ (hence this equality cannot hold). This tells us that $\mathrm{Re}[M_n^R] \geq 0$.

Next in the case that $\mathrm{Re}[M_n^R] = 0$, i.e., $|e^{-M_n^R}| = 1$, the eigensolution in Eq. (18) is extended, which may be satisfied only when either $\beta = 0$ (PBC) or $t_1 = \pm t_{-1}$ (Hermitian or anti-Hermitian). To check this, we denote $e^{-M_n^R(N-2)} = e^{iA}$ and $e^{-M_n^R N} = e^{iB}$, with real numbers $A$ and $B$. Thus Eq. (19) can be expressed as

$$\frac{e^{-\beta} \frac{t_1}{t_{-1}} + e^{iA}}{e^{-\beta} e^{iA} + \frac{t_1}{t_{-1}}} = e^{iB}. \quad (20)$$

Taking modulus square on both sides of Eq. (20), we find that

$$\left| e^{-\beta} \frac{t_1}{t_{-1}} + e^{iA} \right|^2 = \left| e^{-\beta} e^{iA} + \frac{t_1}{t_{-1}} \right|^2. \quad (21)$$

Thus, we arrive at

$$\left[ e^{-2\beta} - 1 \right] \left[ \left( \frac{t_1}{t_{-1}} \right)^2 - 1 \right] = 0, \quad (22)$$

leading to the above mentioned conclusion.

At last it is only possible with $\mathrm{Re}[M_n^R] > 0$, i.e. $|e^{-M_n^R}| < 1$, such that the eigensolutions decay from $x = 1$ to $x = N$. Hence, $e^{-M_n^R N}$ is vanishing and we may drop the higher-order infinitesimal in Eq. (19) to arrive at

$$e^{-M_n^R N} \approx \frac{t_1}{t_1 - t_{-1}} e^{-\beta} \equiv e^{\ln\left(\frac{t_1}{\delta_t}\right) - \beta}, \quad (23)$$

where $\delta_t = t_1 - t_{-1} > 0$ and $N - 2 \approx N$ is taken as $N \gg 1$. Then the decaying exponent $M_n^R$ is given by

$$M_n^R = \frac{\beta - \ln\left(\frac{t_1}{\delta_t}\right)}{N} - \frac{i2n\pi}{N}. \quad (24)$$

Note that the $1/N$ factor is crucial for the quantized response to be discussed later.

The corresponding eigenenergy is given by

$$\begin{aligned} E_n^R &= t_1 e^{-M_n^R} + e^{-\beta} t_{-1} e^{-M_n^R(N-1)} \\ &= t_1 e^{-M_n^R} + e^{-\beta} t_{-1} e^{-\beta + \ln\left(\frac{t_1}{\delta_t}\right)} e^{M_n^R}. \end{aligned} \quad (25)$$

Note again that Eq. (24) does not hold if $\ln\left(\frac{t_1}{\delta_t}\right) > \beta$ because we require $\mathrm{Re}[M_n^R] \geq 0$. Therefore, the condition

$$\delta_t = t_1 - t_{-1} > t_1 / e^\beta. \quad (26)$$

must be satisfied in our consideration and this can be achieved without much difficulty. Indeed, $\delta_t \approx 0$ and $\beta \approx 0$ are close to the Hermitian and PBC limit, respectively, and the above exponentially decaying eigensolutions no longer hold. Nevertheless, our numerical results show a clear quantized response in a wide range of parameters, indicating an intrinsic topological property of the system.

To further illustrate how the eigensolutions lead to the quantized response as discussed in the main text, we assume $t_1 \gg t_{-1}$, i.e. $\delta_t \approx t_1$, so that $\ln(t_1/\delta_t) \approx 0$ and we have

$$t_1 e^{-M_n^R} = E_n^R, \quad M_n^R = \frac{\beta - i2n\pi}{N}. \quad (27)$$

This assumption is mainly for conceptual simplicity. The second term in the eigenenergy expression of Eq. (25) can be neglected as we are working in a parameter regime away from the PBC limit, i.e., $\beta \gg 0$, so that we can use the above exponentially decaying eigensolutions. The following discussion is equally valid for $\beta \to \bar{\beta} \equiv \beta - \ln(t_1/\delta_t)$, as long as the relation in Eq. (26) is satisfied. Note that here $|E_n^R / t_1|$ is determined by the ratio $\beta/N$. That is, the eigenvalues $E_n^R$ will be distributed on a circle on the complex plane, whose radius depends only on $t_1$ as well as the ratio $\beta/N$.

Likewise, the left eigenstates under the same assumptions satisfy $H_\beta^\dagger \left| \Psi_n^L \right\rangle = E_n^L \left| \Psi_n^L \right\rangle$ and $E_n^L = (E_n^R)^*$, and they are found to be

$$\begin{aligned} \psi_{x,n}^L &= C_n^* e^{-M_n^L(N-x)}, \quad t_1 e^{-M_n^L} = E_n^L, \\ M_n^L &= \frac{\beta + i2n\pi}{N}. \end{aligned} \quad (28)$$

From the biorthogonal condition $\left\langle \Psi_n^L | \Psi_n^R \right\rangle = 1$, we then obtain the normalization constant

$$C_n = \frac{e^{-\frac{iN-1}{N}\pi n}}{\sqrt{N e^{-\beta(N-1)/N}}}. \quad (29)$$

With preparations above one important matrix element of the Green's function $G$ can then be found as follows:

$$\begin{aligned} G_{1N} &= \sum_n \frac{1}{E_r - E_n^R} \psi_{1,n}^R \left( \psi_{N,n}^L \right)^* \\ &= \sum_n \frac{1}{E_r - t_1 e^{-\frac{\beta - i2n\pi}{N}}} \frac{1}{N \left( e^{-\frac{\beta - i2n\pi}{N}} \right)^{N-1}} \end{aligned} \quad (30)$$

$$= \sum_n \frac{e^\beta e^{-\frac{\beta - i2n\pi}{N}}}{E_r - t_1 e^{-\frac{\beta - i2n\pi}{N}}} \frac{1}{N}. \quad (31)$$

We may now attempt to rewrite the discrete sum in Eq. (31) in terms of a loop integral of a complex variable $z$, by defining $k_n := 2n\pi/N$ and $z := e^{-\beta/N} e^{ik_n}$. To that end one must implicitly assume that $N$ under consideration is sufficiently large so as to use an integral to replace the discrete sum. With this in mind, for a given $\beta$, $\beta/N$ is assumed to be vanishingly small, and hence essentially we are working in the regime of $|z| \to 1$. Under these conditions, the sum in Eq. (31) can then be evaluated by the following integral

$$G_{1N} = \oint_{|z|=e^{-\beta/N}} \frac{1}{2\pi i} \frac{1}{E_r - t_1 z} \frac{e^\beta}{} dz, \quad (32)$$

which is found to be

$$G_{1N} = -\frac{e^\beta}{t_1}, \quad (33)$$

if $z_0 \equiv E_r / t_1$ satisfies $|z_0| < e^{-\beta/N}$, i.e., the pole of the integrand falls within the integral loop. This condition leads to

$$|E_r| < t_1 e^{-\beta/N} = |E_n^R|, \quad (34)$$

meaning that the reference energy $E_r$ falls within the loop spectrum of $H_\beta$. The above detailed theoretical considerations indicate that, so long as $E_r$ is enclosed by the loop spectrum of $H_\beta$, we have

$$\frac{d \ln |G_{1N}|}{d\beta} = 1, \quad (35)$$

which is just the claim in the main text regarding how to use a quantized physical response to detect the spectral winding number $\nu(E_r)$, as computationally verified in Fig. 5b. The above integral also leads to $G_{1N} = 0$ for $E_r$ outside the loop spectrum of $H_\beta$, namely a signal enters from site $N$ shall vanish at site 1 when the system is

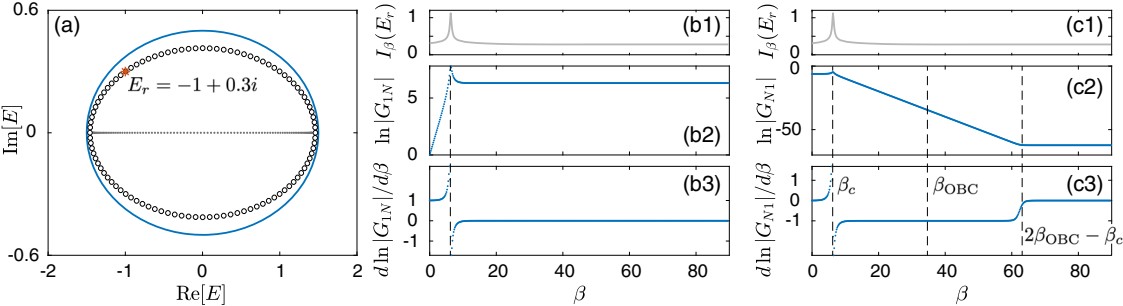

**Fig. 5 The Hatano-Nelson model with only the nearest-neighbor hoppings. a** Spectra under periodic (cyan loop) and open (gray dots) conditions (PBC and OBC), and that within the PBC-OBC interpolation through a modification factor $e^{-\beta}$ of the couplings across the boundary (black circles). The red star indicates the reference energy $E_r = \omega + i\gamma$ for further results in (**b**) and (**c**). Here $E_r$ falls right on the spectrum of the Hamiltonian $H_\beta$, and the value of $\beta = \beta_c \approx 6.2$ is read out from (**b**) and (**c**). **b** and **c** show the amplification ratios given by the off-diagonal elements of the Green's function $|G_{1N}|$ and $|G_{N1}|$ [(**b2**) and (**c2**)] and the derivatives of their logarithms over $\beta$ [(**b3**) and (**c3**)] as functions of $\beta$, where the quantity $I_\beta(E_r)$ indicates with its peaks when $E_r$ is passed through by the $H_\beta$ spectrum. The parameters are $t_1 = 1$ and $t_{-1} = 0.5$.

far away from the PBC limit ($\beta \to 0$). Approaching the PBC limit, the system becomes translational invariant and sites 1 and $N$ are actually two neighboring sites, $G_{1N}$ then must be nonzero. Overall, $|G_{1N}|$ decreases when gradually turning off the boundary hopping by increasing $\beta$, leading to a negative value of $d \ln |G_{1N}|/d\beta$ [as shown in Fig. 3 in the main text].

Next we investigate what happens if we let $\beta$ exceed $\beta_c$, the critical $\beta$ value for which a reference energy point $E_r$ falls exactly on the loop spectrum of $H_\beta$ (as shown in Fig. 5a). Let us first recall the result from Eq. (27), which indicates that the radius of the loop spectrum of $H_\beta$ scales with $\beta/N$. Specifically, for a given lattice size $N$ and a given reference point $E_r$ under investigation, one immediately obtains that

$$\beta_c = -N \ln |E_r/t_1|, \tag{36}$$

which is clearly proportional to $N$. As such, to probe the regime of $\beta \geq \beta_c$, $\beta$ should at least linearly increase with $N$ as well. This being the case, we can no longer approximate the discrete sum in Eq. (31) as a loop integral with $N \to \infty$ in mind, simply because it has the factor $e^\beta$ in its numerator, which diverges with $N$ because in the regime of interest $\beta$ diverges with an increasing $N$.

Based on the discussions above, in the regime of $\beta \geq \beta_c$, what is under investigation becomes the evaluation of the same discrete sum, but with $\beta$ scaling proportionally with $N$, and hence both becoming sufficiently large in the event of using a loop integral to replace this discrete sum. Although we still use the same complex variable $z = e^{-\beta/N} e^{ik_n}$ to invoke a possible loop integral, we see that $|z|$ is now far from unity. To reflect this, we now use the alternative expression of the discrete sum in Eq. (63) and then rewriting it as the following integral, with the integrand having one pole of order $N$ at $z = 0$, and another 1st-order pole at $z = E_r/t_1$,

$$G_{1N} = \oint_{|z|=e^{-\beta/N}} \frac{1}{2\pi i} \frac{1}{z^N(E_r - t_1 z)} dz. \tag{37}$$

Assuming that $|z_0| = |E_r/t_1| > e^{-\beta/N}$, namely, the reference energy $E_r$ is outside the loop, so $z = 0$ is the only pole of the integrand enclosed by the integration path, we have

$$G_{1N} = \frac{t_1^{N-1}}{E_r^N}, \tag{38}$$

a value independent of $\beta$, which is again consistent with the computational results in Fig. 5b2. Further, using the previous expression of $\beta_c$ from Eq. (36), we find that in this case $G_{1N} = \frac{t_1^{N-1}}{E_r^N} = e^{\beta_c}/t_1$. Interestingly, although this magnitude $e^\beta/t_1$ exponentially larger than $e^\beta/t_1$ obtained earlier for $\beta < \beta_c$, this amplification factor is saturated and no longer depends on $\beta$ in the regime of $\beta > \beta_c$.

As a side note, one might wonder why we cannot also use the loop integral in Eq. (37) to treat the first case, namely, a fixed $\beta$ in the regime of $\beta < \beta_c$ but with $N$ approaching sufficiently large values. As said earlier, in this case we essentially perform the summation under the condition of $|z| = 1$. Under this condition we always have $z^N = 1$ and hence the expression in Eq. (37) is no longer useful.

In the same fashion, we can now proceed to examine $G_{N1}$, which depicts how the signal is amplified or suppressed in the other direction. The matrix element $G_{N1}$ is found to be the following,

$$G_{N1} = \sum_n \frac{1}{E_r - E_n^R} \psi_{N,n}^R (\psi_{1,n}^L)^* \tag{39}$$

$$= \sum_n \frac{1}{E_r - t_1 e^{-\frac{\beta - i2\pi n}{N}}} \frac{\left(e^{-\frac{\beta - i2\pi n}{N}}\right)^{N-1}}{N}$$

$$= \sum_n \frac{e^{-\beta} e^{\frac{\beta - i2\pi n}{N}}}{E_r - t_1 e^{-\frac{\beta - i2\pi n}{N}}} \frac{1}{N}. \tag{40}$$

Rewriting Eq. (40) in terms of a loop integral, we have

$$G_{N1} = \oint_{|z|=e^{-\beta/N}} \frac{1}{2\pi i} \frac{e^{-\beta}}{z^2(E_r - t_1 z)} dz. \tag{41}$$

The integrand has a 1st-order pole at $z_0 = E_r/t_1$, and a second-order pole at $z_1 = 0$. Therefore we have

$$G_{N1} = \frac{e^{-\beta} t_1}{E_r^2} \tag{42}$$

if $|E_r| > t_1 e^{-\beta/N}$ (reference energy falls outside the loop of integral), and

$$G_{N1} = 0 \tag{43}$$

if $|E_r| < t_1 e^{-\beta/N}$ (reference energy $E_r$ falls inside the loop of integral). Here because of the factor $e^{-\beta}$ in Eqs. (40) and Eq. (41), the replacement of the discrete sum by the loop integral is always valid by assuming a sufficiently large $N$, i.e., regardless of whether $\beta$ is assumed to be fixed or assumed to scale linearly with $N$. Thus, results obtained above for both $\beta > \beta_c$ and $\beta < \beta_c$ are valid, which are indeed consistent with our numerical results. Note also that for fixed $\beta$, our numerical results for finite systems give a small but nonzero $G_{N1}$ when $\beta < \beta_c$ (e.g. $|G_{N1}| \approx e^{-5}$ in Fig. 5c2), which vanishes when further increasing $N$ (not shown).

Overall, we obtain that the gradient of $\ln |G_{N1}|$ with respect to $\beta$ is again quantized, with

$$\frac{d \ln |G_{N1}|}{d\beta} = -1, \tag{44}$$

if $E_r$ is NOT enclosed by the loop spectrum of $H_\beta$, corresponding to $\beta > \beta_c$ [Fig. 5c]. If we further increase $\beta$ the system shall approach the OBC limit when $\beta = \beta_{OBC} \approx \alpha N$ with $\alpha = \ln(\sqrt{t_1/t_{-1}})$, where the spectrum falls on the same lines as the OBC spectrum[16,53]. Nevertheless, this limit is still not exactly like the OBCs, as the two boundaries are still weakly connected. For example, due to the (weak) boundary couplings, a flux threading cannot be gauged away and can lead to fluctuation of eigenenergies, unlike in real OBC cases. Indeed, from our numerical results, we indeed see that $G_{N1}$ keeps decreasing after $\beta$ exceeds $\beta_{OBC}$, and becomes a constant when $\beta \gtrsim 2\beta_{OBC} - \beta_c$ [Fig. 5c3].

*Cases with only mth-nearest-neighbor coupling.* Consider now a 1D non-Hermitian chain with only the *m*th-nearest-neighbor couplings:

$$H_\beta = \sum_{x=1}^{N-m} (t_m \hat{c}_x^\dagger \hat{c}_{x+m} + t_{-m} \hat{c}_{x+m}^\dagger \hat{c}_x)$$
$$+ e^{-\beta} \sum_{x=N-m+1}^{N} (t_m \hat{c}_x^\dagger \hat{c}_{x+m-N} + t_{-m} \hat{c}_{x+m-N}^\dagger \hat{c}_x). \tag{45}$$

Here we first assume $N/m$ is an integer, thus the system is decoupled into $m$ identical 1D subchains, and the eigenstates satisfy

$$t_m \psi_{x+m,n}^R + t_{-m} \psi_{x-m,n}^R = E_n^R \psi_{x,n}^R \tag{46}$$

for $x \in [2, N-m]$, and

$$e^{-\beta} t_m \psi_{s,n}^R + t_{-m} \psi_{N-2m+s,n}^R = E_n^R \psi_{N-m+s,n}^R, \tag{47}$$

$$t_m \psi_{m+s,n}^R + e^{-\beta} t_{-m} \psi_{N-m+s,n}^R = E_n^R \psi_{s,n}^R, \tag{48}$$

with $s = 1, 2, \ldots, m$ labelling different subchains. As in the previous discussion for the case of $m = 1$, we assume $t_m \gg t_{-m}$ to obtain some simple analytical results.

Here we replace the labels $x$ and $n$ with $x_s$ and $n_s$ for each subchain. We take the ansatz

$$\psi^R_{x_s,n_s} = C_{n_s} e^{-M^R_{n_s}(x_s-1)}, \qquad (49)$$

with $n_s$ only takeing values from 1 to $N_m = N/m$, given that each subchain contains only $N_m$ lattice sites, and $x_s = (x-s+m)/m$ ranging from 1 to $N_m$, with $x$ being $s, m+s, 2m+s, \ldots, N-m+s$, and more importantly,

$$t_m e^{-M^R_{n_s}} = E^R_{n_s}, M^R_{n_s} = \frac{\beta - i2n_s\pi}{N_m}. \qquad (50)$$

Similarly, the left eigenstates are given by

$$\psi^L_{x_s,n_s} = C^*_{n_s} e^{-M^L_{n_s}(N_m-x_s)}, t_m e^{-M^L_{n_s}} = E^L_{n_s},$$
$$M^L_{n_s} = \frac{\beta + i2n_s\pi}{N_m}. \qquad (51)$$

Again, from the biothorgonal condition $\langle \Psi^L_n | \Psi^R_n \rangle = 1$, we have the normalization constants

$$C_{n_s} = \frac{e^{-i\frac{N_m-1}{N_m}\pi n_s}}{\sqrt{N_m e^{-\beta(N_m-1)/N_m}}}. \qquad (52)$$

Note that in the Green's function matrix, the element $G_{1N}$ shall always be zero as each subchain is decoupled from the others. In this case, the directional amplification of each subchain corresponds to the element $G_{s(N-m+s)}$, with

$$\begin{aligned}
G_{s(N-m+s)} &= \sum_{n_s} \frac{1}{E_r - E^R_{n_s}} \psi^R_{x_s=1,n_s} \left(\psi^L_{x_s=N_m,n_s}\right)^* \\
&= \sum_{n_s} \frac{1}{E_r - t_m e^{\frac{\beta-i2n_s\pi}{N_m}}} \frac{1}{N_m \left(e^{\frac{-i2n_s\pi}{N_m}}\right)^{N_m-1}} \\
&= \sum_{n_s} \frac{e^\beta e^{\frac{-i2n_s\pi}{N_m}}}{E_r - t_m e^{\frac{-i2n_s\pi}{N_m}}} \frac{1}{N_m}.
\end{aligned} \qquad (53,54)$$

As $n_s$ takes value from 1 to $N_m = N/m$, here we need to define $k_s = i2mn_s\pi/N$, so that the summation can be replaced by an integral with $k_s$ varying from 0 to $2\pi$. Similar to the case with only nearest-neighbor coupling, we then have

$$\frac{d \ln|G_{s(N-m+s)}|}{d\beta} = 1 \qquad (55)$$

for each subchain, when $E_r$ is enclosed by the loop-like spectrum of each subchain. This result indicates that each subchain has its own spectral winding number $\nu_s(E_r) = 1$, but for the original 1D chain with $m$th-nearest-neighbor couplings, the element $G_{1N}$ and $G_{N1}$ are zero as sites 1 and $N$ belong to different decoupled subchains.

On the other hand, the $m$-subchain picture here also indicates an effective unit-cell structure with $m$ sublattices, even though the sublattices are physically equivalent on the lattice. Thus the directional amplification of the overall system comprised by these subchains/sublattices shall be described by the combination of that of each subchain, corresponding to the corner blocks of the overall Green function matrix,

$$G_{\leftarrow,m\times m} = \begin{pmatrix} G_{1(N-m+1)} & G_{1(N-m+2)} & \cdots & G_{1N} \\ G_{2(N-m+1)} & G_{2(N-m+2)} & \cdots & G_{2N} \\ \vdots & \vdots & \vdots & \vdots \\ G_{m(N-m+1)} & G_{m(N-m+2)} & \cdots & G_{mN} \end{pmatrix}, \qquad (56)$$

$$G_{\rightarrow,m\times m} = \begin{pmatrix} G_{(N-m+1)1} & G_{(N-m+1)2} & \cdots & G_{(N-m+1)m} \\ G_{(N-m+2)1} & G_{(N-m+2)2} & \cdots & G_{(N-m+2)m} \\ \vdots & \vdots & \vdots & \vdots \\ G_{N1} & G_{N2} & \cdots & G_{Nm} \end{pmatrix}, \qquad (57)$$

for signals moving toward the left and the right side, respectively. In the above schematic scenario where the subchains are fully decoupled from each other, only the diagonal elements of the above two matrices are nonzero, i.e., $(G_{\leftarrow,m\times m})_{ab} \approx -\delta_{ab}e^\beta/t_1$ (see Eq. (33)) when $E_r$ is enclosed by the spectrum on the complex plane, and $(G_{\rightarrow,m\times m})_{ab} \approx \delta_{ab}e^{-\beta}t_1/E_r^2$ (see Eq. (42)) when $E_r$ is NOT enclosed by the spectrum on the complex plane. This being the case, we arrive at

$$\det[G_{\leftarrow,m\times m}] \propto e^{m\beta}, \ \det[G_{\rightarrow,m\times m}] \propto e^{-m\beta}, \qquad (58)$$

in the above two cases, repectively. Thus we have

$$\nu_{\leftarrow,m} := \frac{d\ln|G_{\leftarrow,m\times m}|}{d\beta} = m, \qquad (59)$$

corresponding to the spectral winding number $\nu(E_r) = m$ for the case with only the $m$th-nearest-neighbor couplings. Furthermore, this conclusion shall also be valid when

$N/m$ is not an integer. In such cases, the system still possesses the $m$-subchain picture, only that the two ends of one subchain are connected to those of other subchains now. This conclusion is also verified by our numerical calculations. The independent subchain picture developed here offers an important method to understand more realistic situations where couplings with different hopping ranges coexist.

*Cases with couplings across multiple ranges.* In the main text, using the concept of counting the number of effectively independent modes whose amplification factor has the $e^\beta$ dependence, we have physically explained why the determinant of a subblock of the Green's function can still be used to capture the spectral winding number. Here we use another method to illuminate on the origin of classical quantized response. Let us assume that the largest hopping distance is $m$ lattice sites. We first assume that the system's hopping is still dominated by $m$th-nearest-neighbor hopping so that the previous $m$-subchain picture applies. We write the equation of the Green's function in the following form

$$G = \begin{pmatrix} G_{m\times(N-m)} & G_{\leftarrow,m\times m} \\ \vdots & \vdots \end{pmatrix}, \qquad (60)$$

$$E_r - H = \begin{pmatrix} H^r_{(N-m)\times m} & \cdots \\ H^r_{m\times m} & \cdots \end{pmatrix}, \qquad (61)$$

$$G_{m\times(N-m)}H^r_{(N-m)\times m} + G_{\leftarrow,m\times m}H^r_{m\times m} = \mathbb{I}_{m\times m}, \qquad (62)$$

with $G_{m\times(N-m)}$, $H^r_{(N-m)\times m}$, and $H^r_{m\times m}$ being different blocks of the $G$ and $E_r - H$ matrices, whose sizes are indicated by their subscripts. Note that for the two blocks of $H^r = E_r - H$, the coefficient $e^{-\beta}$ is only contained in the block $H^r_{m\times m}$. On the other hand, $H^r_{(N-m)\times m}$ has nonzero and $\beta$-independent elements only in the first $2m$ rows. Consider then the first $2m$ columns of $G_{m\times(N-m)}$ in Eq. (62). Using the $m$-subchain picture as the starting point of consideration, the explicit expressions of $G_{m\times(N-m)}$ are obtained as the following,

$$\begin{aligned}
G_{xy} &= \sum_n \frac{1}{E_r - E^R_n} \psi^R_{x,n} \left(\psi^L_{y,n}\right)^* \\
&= \sum_n \frac{1}{E_r - t_1 e^{\frac{\beta-i2n\pi}{N}}} \frac{e^{-\frac{\beta-i2n\pi}{N}(N-y+x-1)}}{N\left(e^{\frac{\beta-i2n\pi}{N}}\right)^{N-1}} \\
&= \sum_n \frac{e^{-\frac{\beta-i2n\pi}{N}(x-y)}}{E_r - t_1 e^{\frac{\beta-i2n\pi}{N}}} \frac{1}{N},
\end{aligned} \qquad (63,64)$$

which yields an integral expression

$$G_{xy} = \oint_{|z|=e^{-\beta/N}} \frac{1}{2\pi i} \frac{z^{x-y-1}}{E_r - t_1 z} dz. \qquad (65)$$

This integral expression does not depend on $\beta$. With these preparation and now inspecting Eq. (62) again, it is seen that $G_{\leftarrow,m\times m}$ should have a $e^\beta$ coefficient, so as to cancel the $e^{-\beta}$ coefficient in the matrix $H^r_{m\times m}$. Therefore the determinant of $G_{\leftarrow,m\times m}$ yields a coefficient of $e^{m\beta}$, and can hence reflect the spectral winding number $\nu(E_r) = m$.

Let us now gradually strengthen all other hoppings with different length scales. Before the spectral loop hits the reference energy $E_r$, $G$ is well defined, the above arguments are expected to hold and hence the response plateau is quantized at $m$, thus indicating the topological robustness of the quantized response. In other words, this current picture breaks down when the spectral loop passes the reference energy (where $(E_r - H)$ has zero eigenvalues), during which the qualitative nature of the Green's function changes drastically. In particular, after the phase transition the system has a smaller spectral winding number, say $\nu(E_r) = m - 1$ for example, and hence the system is now topologically equivalent to a simpler case with $m - 1$ subchains. One should then use $(m - 1)$-subchain scenario to reapply the above insight self-consistently, by considering the block $G_{\leftarrow,(m-1)\times(m-1)}$ instead to measure the topological winding number $m - 1$. This is indeed confirmed in our extensive numerical results for several systems with different spectral winding numbers.

## Data availability
Raw numerical data from the plots presented are available from the authors upon request.

## Code availability
Although not essential to the central conclusions of this work, computer codes for generating our figures are available from L.L. and C.H.L. upon reasonable request.

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

## Acknowledgements

We would like to thank Da-Jian Zhang for helpful discussions. L.L. acknowledges funding support by the Key-Area Research and Development Program of GuangDong Province under Grants No. 2019B03033000, and the Guangdong Basic and Applied Basic Research Foundation (2020A1515110773). J.G. acknowledges funding support by the Singapore NRF Grant No. NRF-NRFI2017-04 (WBS No. R-144-000-378-281). C.H.L. acknowledges funding support by the Singapore MOE Tier-1 start-up grant (WBS R-144-000-435-133).

## Author contributions

L.L. carried out preliminary studies and all authors participated in the discussions. S.M. helped to improve the analysis of the high-spectral-winding scenario. C.H.L. proposed the physical understanding of classical quantization and refined this project extensively. L.L. and C.H.L. carried out additional theoretical and computational studies. All authors discussed the results and participated in the writing of the manuscript. J.G. finalized the manuscript.

## Competing interests

The authors declare no competing interests.
