## [Peer Review File · Nature Communications]

REVIEWER COMMENTS

Reviewer #1 (Remarks to the Author):

The main result of the paper is a theoretical proposal for measuring the spectral winding number in classical setups (electric circuits for example). The authors propose to measure the winding number by measuring the two-point Green's function at the open boundaries. They conjecture that the exponent of the Green's function is proportional to the winding number. So far the proposal is restricted to one-band in 1D.

The authors connect the Green function with spectral winding number, which makes the change of spectral winding observable in electric circuit experiments. Specifically, the derivative of the logarithm of Green function with respect to parameter β versus β has quantized plateaus under thermodynamic limit, and the first quantized plateau corresponds to the winding number with respect to the reference energy E_r . Here β is a parameter that controls the boundary condition from periodic to open boundary.

The spectral winding number is a new topological invariant unique to non-hermitian systems. In 1D, this is the winding of the phase of the eigenvalue over the BZ, with reference to a pre-chosen energy, and this number being number has been shown in previous works related to the skin effect. It would be nice if one finds a more quantitative "bulk-edge correspondence" for this topological invariant, which I believe is the goal of this paper. Unfortunately, after reading it, I find the derivation of the paper too restricted for its claim. Therefore, I cannot recommend recommendation unless the following questions be addressed.

1. The authors propose the quantized response to measure the spectral winding in one-band cases. For completeness, I think that the authors should extend this statement into multi-band cases. In multi-band cases, the spectral winding for a given reference energy is well-defined and the parameter β can also tune the boundary condition from periodic to open. Based on this, how to define the quantized response to capture the spectral topology? More concretely, what the form of Eq.(10) will become in multi-band cases? Can the authors give certain numerical results to support it?
2. The key step to obtain the quantized response is that the (determinant of) Green's function is proportional to the exponential of $m\beta$. I note that it has been proved for the cases with only nearest neighbor coupling and only m th-nearest neighbor coupling in method section. However, to obtain the final result Eq.(27), " $t_1 \gg t_{-1}$ " is assumed, where the system approaches to an exceptional point. In general cases with only nearest neighbor coupling, the magnitudes of t_1 and t_{-1} are comparable, can the same conclusion Eq.(29) be strictly obtained at this time?
3. For general one-band cases with more complicated hopping parameters, the authors take a "sub-chain" argument, which is strictly established for the cases with only m th-nearest neighbor coupling in method section, and further obtain Eq.(10). However, in general cases, the first $(m-1)$ th hopping parameters are not zero and even dominated, we expect that the authors can give a more rigorous and clear proof instead of only the "sub-chain" argument.
4. If the system has disorders, the translational symmetry is destroyed, and the spectral winding in momentum space will be not well-defined. In these cases, can the quantized plateaus still be observed?
5. In addition, I suggest that the content in the section "Classical vs. Quantum response", especially the formulas, can be partially moved to the Supplementary Materials for the consideration that they are not very closely related to the focus of the main text, and these formulas can also be found in

most textbooks.

Reviewer #2 (Remarks to the Author):

This manuscript presents a theoretical study on the quantized responses associated with the spectral winding number of 1D non-Hermitian Hamiltonians in a classical setting in the sense that it is based on the Green's function of the system. In the model Hamiltonian, the coupling between the first and last sites is tuned by a scaling factor β such that the boundary condition continuously changes from periodic (PBC) to open (OBC). During this process, the winding number reduces to zero as the complex eigenenergy deforms from loops in PBC to lines in OBC that do not have interiors. The authors further show that the logarithm of the Green's function describing directional signal amplification shows plateaus with slopes related to the winding number, and exhibits jumps as the winding number decreases during the PBC to OBC transition. The authors argue that the quantized response of the logarithm of the Green's function is a measurable quantity and show an example of measuring the quantized impedance in an electrical circuit setting.

Overall, I think that the proposal of using the Green's function to measure the winding number is new and interesting, and the presented results appear sound. I believe this work provides new insights and makes an important contribution to the field of non-Hermitian physics. However, the manuscript is focused on the detailed physics of an already well-studied model system. I am not convinced that it is significant enough to warrant publication in Nature Communications.

I also have a few comments regarding the details of the manuscript.

1. As far as I understand, the winding number should be a well-defined invariant for a 1D non-Hermitian Hamiltonian with respect to an arbitrary reference energy E_r . It seems that the system in the manuscript can have different winding numbers as the reference energy E_r is chosen differently. Physically how does one choose the reference energy? Do the results in Figs. 2 and 3 still hold if one chooses the reference E_r differently to start with?
2. In the Method section, the assumption of calculating G_{1N} using the integral form in Eq. 26 is that β/N is vanishingly small. Yet in the main text, with $N = 100$, the β parameter is varied from 0 to 90, which is beyond this assumption. Does the result of G_{1N} calculated numerically by Eq. 25 deviate from Eq. 26 across the wide range of β ?
3. On page 3, the authors claim a quantized classical response distinct from existing topological Hall response exclusively in quantum systems. However, this statement completely ignores the abundant examples of various Hall responses realized in photonic and sonic systems which are also classical.
4. The inverse temperature and the scaling factor of boundary couplings are both denoted by β , which causes confusion.
5. The explanation regarding winding number of -1 for the topological trivial case on page 5 seems not provided in the Supplemental Material.

First of all, we would like to thank you and the reviewers for taking your time in considering our manuscript. We are pleased that both reviewers found our work interesting, and unanimously recognized the novelty of our work in the very active field of non-Hermitian physics. Reviewer #1 acknowledge that “it would be nice if one finds a more quantitative ‘bulk-edge correspondence’ for this topological invariant”, His/her reservation is only that our derivation is too restricted to a special type of model, which we have greatly improved upon with the suggestions from the reviewer. Reviewer #2 agrees our work “is new and interesting” and “provides new insights and makes an important contribution to the field of non-Hermitian physics”.

By following the detailed comments from both referees closely, we have substantially improved on our manuscript by greatly enlarging the scope of our results. We are confident that we have now fully addressed all their concerns, such that our work now meets the standards of Nature Communications. In this revised manuscript, we have broadened the scope of our results and derivations to also include:

- 1) A treatment of generic multi-band scenarios
- 2) A treatment of models with generic longer-range hoppings, beyond the “sub-chain” argument
- 3) A demonstration that disorder does not change the main results
- 4) More involved treatment of the topological invariance of our classical response

These enhancements are performed closely according to the suggestions from both referees, and definitely push the scope of our results to far beyond the detailed physics of particular well-studied model systems.

Detailed below are our responses to the comments of the reviewers, together with a description of the associated changes in the manuscript.

Response to Reviewer #1

The main result of the paper is a theoretical proposal for measuring the spectral winding number in classical setups (electric circuits for example). The authors propose to measure the winding number by measuring the two-point Green's function at the open boundaries. They conjecture that the exponent of the Green's function is proportional to the winding number. So far the proposal is restricted to one-band in 1D.

The authors connect the Green function with spectral winding number, which makes the change of spectral winding observable in electric circuit experiments. Specifically, the derivative of the logarithm of Green function with respect to parameter β versus β has quantized plateaus under thermodynamic limit, and the first quantized plateau corresponds to the winding number with respect to the reference energy E_r . Here β is a parameter that controls the boundary condition from periodic to open boundary.

The spectral winding number is a new topological invariant unique to non-hermitian systems. In 1D, this is the winding of the phase of the eigenvalue over the BZ, with reference to a pre-chosen energy, and this number being number has been shown in previous works related to the skin effect. It would be nice if one finds a more quantitative "bulk-edge correspondence" for this topological invariant, which I believe is the goal of this paper. Unfortunately, after reading it, I find the derivation of the paper too restricted for its claim. Therefore, I cannot recommend recommendation unless the following questions be addressed.

We are pleased that Reviewer #1 acknowledges the importance of our finding, namely a more quantitative “bulk-edge correspondence” of the spectral winding number unique to non-Hermitian systems. Now that we have significantly generalized our derivations and results, as closely according to the suggestions of the reviewer, we are confident that our work is no longer “too restrictive” for claiming this topological correspondence. We are grateful to him/her for the following suggestions for improvement, which help us broadening the scope of our finding.

1. The authors propose the quantized response to measure the spectral winding in one-band cases. For completeness, I think that the authors should extend this statement into multi-band cases. In multi-band cases, the spectral winding for a given reference energy is well-defined and the parameter β can also tune the boundary condition from periodic to open. Based on this, how to define the quantized response to capture the spectral topology? More concretely, what the form of Eq.(10) will become in multi-band cases? Can the authors give certain numerical results to support it?

We are grateful to our reviewer for raising this insightful question. It is indeed interesting to extend our finding to multi-band cases, as now different bands may have different spectral winding. Another subtly is the following: in this extended case how shall we take into account the unit-cell structure when we choose the size of the block of the Green’s function matrix.

To this end, we propose to measure the winding based on a $m \times m$ block of the Green’s function matrix with only a single sublattice included, with m equaling the total spectral winding number for a chosen reference energy, generalizing Eq 10. This is because by choosing a single sublattice with non-zero support of the steady state, it already contains sufficient information from one unit cell to another unit cell.

We have numerically verified this conclusion with an extended non-Hermitian Su-Schrieffer-Heeger model, with several intriguing cases in which the total spectral winding number for both bands take 1, 2, and 3. In some cases the two bands even connect into one big loop, thus each band corresponding to a fractional winding number. Yet our scheme is seen to work well. We have discussed these results in the new Supplemental Note 4, which is also summarized in the main text.

As such, we have successfully shown how to generalize our approach to multi-band settings, and computationally verified it under a variety of scenarios. The newly added results definitely make our main claim more general and more fascinating, for which we are grateful to our referee for having stretched our efforts on this part.

2. The key step to obtain the quantized response is that the (determinant of) Green's function is proportional to the exponential of $m\beta$. I note that it has been proved for the cases with only nearest neighbor coupling and only m -nearest neighbor coupling in method section. However, to obtain the final result Eq.(27), " $t_1 \gg t_{-1}$ " is assumed, where the system approaches to an exceptional point. In general cases with only nearest neighbor coupling, the magnitudes of t_1 and t_{-1} are comparable, can the same conclusion Eq.(29) be strictly obtained at this time?

We thank our reviewer for his/her careful reading of our manuscript and brought up this valuable question. Previously, the assumption " $t_1 \gg t_{-1}$ " was made for simplicity of presentation, and our results and derivations definitely hold true for for generic $t_1 > t_{-1}$ " with no other constrain on relative magnitude.

In the revised version, we have made this explicit by rewriting the derivation without the assumption " $t_1 \gg t_{-1}$ ", with comments on the new validity conditions. This derivation only requires a much weaker condition on t_1 and t_{-1} , which is also associated with the parameter β , and can be far away from the exceptional point (i.e. $t_{-1}=0$). The revised discussion is marked in blue in the Methods section.

3. For general one-band cases with more complicated hopping parameters, the authors take a "sub-chain" argument, which is strictly established for the cases with only m -nearest neighbor coupling in method section, and further obtain Eq.(10). However, in general cases, the first $(m-1)$ th hopping parameters are not zero and even dominated, we expect that the authors can give a more rigorous and clear proof instead of only the "sub-chain" argument.

Being able to account for cases beyond the simple m -th nearest neighbor coupling is a vital aspect of our discovered quantized response. It is fair to say that precisely because the quantized response can cover more complicated cases (even covering the multi-band cases we discussed earlier in this reply), we are making a fascinating advance in investigating physics. We also perfectly understand our referee's wish to see a "rigorous" proof, and below we share with our referee what is the key physics lessons learned that could point to a future *mathematical* proof, which is still out of reach at

this point. In our view, physical understanding and actual demonstration can be even more interesting than a technical proof as we attempt to break a new ground.

We start with a case with the m -th nearest neighbor hopping only. In this case, the whole system can be regarded as m decoupled chains with identical nearest neighbor hopping. That these chains are independent is reflected by the fact that the $m \times m$ Green's function block is an entirely diagonal matrix. If the reference energy hits the spectral loop for one chain, so do all other chains. As shown rigorously in the Methods, once the reference energy is outside the spectral loop, there will be either no amplification or saturated amplification. So the quantized winding “ m ” or “ 0 ” in this case counts the number of modes that can be amplified, with each mode having an exponential dependence on β .

Now if we turn on coupling of different hopping ranges (all smaller than or equal to m), the system becomes m sub-chains coupled to each other. So how to correctly count the number of independent modes that can be amplified, still with their amplification factor exponentially dependent on β ? The obvious and physics-motivated way is to diagonalize the $m \times m$ sub-block of the Green's function to extract effectively independent modes that is still amplified with the amplification factor e^{β} . In particular, this key information can be captured by the determinant of the $m \times m$ block, which is nothing but the product of all the eigenvalues of the sub-block matrix of the Green's function.

We can now consider two scenarios to see what happens if we move the system across a phase transition. Consider the first scenario where we tune β . As shown rigorously in the Methods, if the spectral loop encloses the reference energy, the amplification factor of a simple chain with only nearest neighbor hopping will scale as $\exp\{\beta\}$. Once the β -dependent spectral loop shrinks, crosses, and eventually does not enclose a given reference energy, then the amplification will be saturated and hence no longer dependent on β . This process will be experienced by one of the m effectively independent sub-chains when a phase transition occurs. The rest $(m-1)$ independent modes of the $m \times m$ Green's block will remain amplified, with their amplification factor still exponentially dependent on β . Therefore the log of the determinant of this $m \times m$ Green's block will be quantized at $m-1$. As we further tune β , we should be able to gradually experience quantized jumps towards to lower and lower integer values, simply because the number of independent sub-chains that can still amplify signal this way will decrease one by one. This is also verified by our computational findings again and again.

In the second scenario, we do not tune β but only varying other system parameters, such as the reference energy or hopping constants. Applying the above decoupled mode picture again, one knows that one of the independent modes will first cease to amplify when a phase transition occurs, hence it is topologically equivalent to a chain with zero winding. In this case, as we also show explicitly in the main text using the simplest chain, the eigenvalue of this trivial mode can have β -dependence like $\exp\{-C\beta\}$. (This point is also related to our reply to our second referee's last comment). In this case, to witness the actual winding number, which is $(m-1)$ after the first phase transition, one should consider instead the determinant of the $(m-1) \times (m-1)$ sub-block of the Green's function. This understanding is also fully accounted for when we show how to choose the right sub-block of the Green's function to map out a phase diagram completely (see Fig 3).

$$\begin{array}{c}
\text{G matrix} \qquad \qquad \text{Er-H matrix} \\
\left(\begin{array}{c} \boxed{\phantom{G_{m \times (N-m)}}} \\ \boxed{G_{m \times (N-m)} \quad G_{\leftarrow, m \times m}} \end{array} \right) \left(\begin{array}{c} \boxed{\phantom{H_{(N-m) \times m}^r}} \\ \boxed{H_{(N-m) \times m}^r} \\ \boxed{\phantom{H_{m \times m}^r}} \\ \boxed{H_{m \times m}^r} \end{array} \right) = \mathbb{I} \\
G_{m \times (N-m)} H_{(N-m) \times m}^r + G_{\leftarrow, m \times m} H_{m \times m}^r = \mathbb{I}_{m \times m} \\
G_{\leftarrow, m \times m} \sim e^\beta, \quad H_{m \times m}^r \sim e^{-\beta}
\end{array}$$

Figure Caption: Properties of matrix elements and the defining equation for the Green's function.

We are sure that our referee may also appreciate a working mechanism based more on technical equations. Let us inspect the defining equation of the Green's function itself (see Figure above). First assuming that the system is essentially that of m subchains of which we have analytical results. From the equation of $G(E_r-H)=1$ (elaborated in detail in the revised Methods section), the $m \times m$ sub-block of the Green's function matrix G we used to observe quantized response must have an overall coefficient e^β , to cancel the coefficient of the matrix elements (E_r-H) that all have a coefficient $e^{-\beta}$ in its corresponding $m \times m$ sub-block (see Figure above). As such, the determinant of $m \times m$ off-diagonal block of G will always have a coefficient of $e^{m\beta}$, provided that G is well defined in the first place. Let us now gradually turn on other hopping parameters. This picture is expected to hold until the reference energy hits the spectral loop, where G is ill-defined. Therefore, the log of the determinant of the $m \times m$ off-diagonal block of G , which is measurable in a signal amplification setting, is quantized at m until a phase transition occurs. Further, this quantization at m is robust due to topology, hence generating a plateau. Self-consistently, upon crossing the phase transition point, the topological winding number jumps, e.g., from m to $m-1$, the system is then topologically equivalent to that of $(m-1)$ simple sub-chains and one should reapply the above analysis with $(m-1)$ sub-chains, obtaining a quantization plateau at $m-1$. This also makes it clear that quantized response is protected by spectral winding topology.

In making a stronger case for our method, we have also decided to add these physics discussions to the Methods section under "Cases with couplings across multiple ranges", which comes after introducing about the sub-chain picture.

4. If the system has disorders, the translational symmetry is destroyed, and the spectral winding in momentum space will be not well-defined. In these cases, can the quantized plateaus still be observed?

This is a very important question as the robustness of the quantized response against disorder is a key aspect for topological quantization. Fortunately, our quantized plateaus turn out to be indeed robust against disorder. Furthermore, as the spectral winding topology of the non-Hermitian

Hamiltonian does not depend on the system's symmetries, we expect the quantized plateaus to be robust against disorder without any symmetry restriction.

In our revision, we have considered reasonably large disorder of the hopping parameters, and found that the quantized plateaus can still be clearly seen. Even with relatively large disorder of maximal magnitude $W=0.5$ (Eq S5), the plateaus are still very distinct, just somewhat rounded (Fig. S3). We have added a new section in the Supplemental Materials (Note 2) to show these affirmative results,

5. In addition, I suggest that the content in the section "Classical vs. Quantum response", especially the formulas, can be partially moved to the Supplementary Materials for the consideration that they are not very closely related to the focus of the main text, and these formulas can also be found in most textbooks.

We have now accordingly reorganized the relevant content as our reviewer had suggested. Many thanks to our referee.

Response to Reviewer #2

Reviewer #2 (Remarks to the Author):

This manuscript presents a theoretical study on the quantized responses associated with the spectral winding number of 1D non-Hermitian Hamiltonians in a classical setting in the sense that it is based on the Green's function of the system. In the model Hamiltonian, the coupling between the first and last sites is tuned by a scaling factor β such that the boundary condition continuously changes from periodic (PBC) to open (OBC). During this process, the winding number reduces to zero as the complex eigenenergy deforms from loops in PBC to lines in OBC that do not have interiors. The authors further show that the logarithm of the Green's function describing directional signal amplification shows plateaus with slopes related to the winding number, and exhibits jumps as the winding number decreases during the PBC to OBC transition. The authors argue that the quantized response of the logarithm of the Green's function is a measurable quantity and show an example of measuring the quantized impedance in an electrical circuit setting.

Overall, I think that the proposal of using the Green's function to measure the winding number is new and interesting, and the presented results appear sound. I believe this work provides new insights and makes an important contribution to the field of non-Hermitian physics. However, the

manuscript is focused on the detailed physics of an already well-studied model system. I am not convinced that it is significant enough to warrant publication in Nature Communications.

We thank our reviewer for his/her positive comment that “this work provides new insights and makes an important contribution to the field of non-Hermitian physics.” The main reservation of our reviewer is that he/she thinks our work is focused on the detailed physics of an already well-studied model system.

We understand where our referee’s concern come from, given that our discussions start from a well-studied single-band model for illustration and for clear physics insights. However, as also stated in response to our reviewer 1, we have now significantly generalized and explained the results to a wide variety of much more generic models (multi-band cases, systems with multi-range hopping, with the same quantization physics). It is also a significant finding by itself that classical measurements of some signals can display quantized plateaus. We would also like to highlight that the spectral winding topology describes a new type of topology unique in non-Hermitian systems, which has not be unveiled until recent years [1-3], and our results on how such spectral winding can be measured through quantized response are certainly timely and motivating.

While there are similarities between the non-Hermitian spectral winding number and the winding number describing conventional 1D topological insulators, they actually describe very different physical properties. In conventional 1D topological insulators, the winding number is defined for the winding of the spin texture throughout the Brillouin zone, and hence requires at least a 2-band Hamiltonian. On the other hand, the spectral winding number is defined as the winding of the complex eigenenergies, and can be nontrivial even for single-band systems (e.g. the starting examples in our manuscript). As a consequence, the physical phenomena related to these two types of topology are also fundamentally different. For example, in conventional topological insulators, the number of topological edge states under OBC has a one-to-one correspondence to the topological invariant. Yet in non-Hermitian systems, a nonzero spectral winding number only predicts the existence of the so-call skin edge modes under OBC [4].

To date, experimental observations of the non-Hermitian spectral topology have already become one of the most frontier directions. Various experimental setups have been used to realize the so-call non-Hermitian bulk-boundary correspondence through the non-Hermitian skin effect [5-8], which however only reflects the existence of a nonzero spectral winding number, but not its exact value. It is not until this year that an observation of arbitrary spectral winding numbers is achieved [9]. Nevertheless, such an observation is based on the reconstruction of the spectrum of the system, after which the spectral winding number is read out manually. By contrast, our proposal here is based on the quantization of certain quantity in a process of signal amplification, which is an analog of the Hall conductance measurement of conventional Hermitian topological insulators. In short, our work provides the first one-to-one correspondence between the spectral winding number with a quantized quantity, which is experimentally measurable through a process of directional signal amplification [10]. Our scheme also unveils a topological quantized response valid not only for quantum systems, but also for classical systems, which shall greatly broaden the field of topological

properties in physical systems. Thus we believe our work is indeed beyond “the detailed physics of an already well-studied model system”, and meets the high standard of Nature Communications.

- [1] H. Shen, B. Zhen, and L. Fu, “Topological Band Theory for Non-Hermitian Hamiltonians”, *Phys. Rev. Lett.* 120, 146402 (2018).
- [2] N. Okuma, K. Kawabata, K. Shiozaki, and M. Sato, “Topological origin of non-hermitian skin effects,” *Phys. Rev. Lett.* 124, 086801 (2020).
- [3] Kai Zhang, Zhesen Yang, and Chen Fang, “Correspondence between winding numbers and skin modes in non-Hermitian systems,” *Phys. Rev. Lett.* 125, 126402 (2020).
- [4] S. Yao and Z. Wang, “Edge states and topological invariants of non-Hermitian systems”, *Phys. Rev. Lett.*, 121, 086803 (2018).
- [5] M. Brandenbourger, et al., “Non-reciprocal robotic metamaterials.” *Nat. Commun.* 10, 4608 (2019).
- [6] L. Xiao, et al., “Non-Hermitian bulk-boundary correspondence in quantum dynamics”, *Nat. Phys.* 16, 761–766 (2020).
- [7] A. Ghatak, et al., “Observation of non-Hermitian topology and its bulk-edge correspondence” *PNAS* 117, 29561-29568 (2020).
- [8] T. Helbig, et al., “Generalized bulk-boundary correspondence in non-Hermitian topoelectrical circuits”, *Nat. Phys.* 16, 747–750 (2020).
- [9] K. Wang, A. Dutt, K. Y. Yang,, C. C. Wojcik, J. Vučković, and S. Fan, “Generating arbitrary topological windings of a non-Hermitian band”, *Science* 371, 1240-1245 (2021).
- [10] C. C. Wanjura, et al., “Topological framework for directional amplification in driven-dissipative cavity arrays,” *Nat. Commun.* 11, 3149 (2020).

I also have a few comments regarding the details of the manuscript.

1. As far as I understand, the winding number should be a well-defined invariant for a 1D non-Hermitian Hamiltonian with respect to an arbitrary reference energy E_r . It seems that the system in the manuscript can have different winding numbers as the reference energy E_r is chosen differently. Physically how does one choose the reference energy? Do the results in Figs. 2 and 3 still hold if one chooses the reference E_r differently to start with?

The answer to this question is yes. Indeed, the reference energy is a tunable parameter of an experimental measurement of the spectral winding topology. Physically, it takes an analogous role as the chemical potential/Fermi energy offset, except that for classical systems, we are much more

free to choose this offset. As described in the subsection “Measurement of quantized response in electrical Circuits”, this E_r , in the context of circuits, can be implemented by suitable adjusting a grounding circuit to the circuit to be measured. In other classical systems i.e. photonic or mechanical systems, we can also attach appropriate on-site resonators or mass-loading mechanisms to simulate E_r .

In Supplemental Note 2, we have illustrated in Fig S1 results that expand upon those of Fig. 2, to a more complicated parameter regime, where the winding number ranges from -1 to 2. Evidently, the quantization plateaus are exactly as predicted by our approach, for different E_r . In addition, Fig. 3(a) has already been plotted by scanning the value of E_r in a large parameter regime, as we have defined $E_r = \omega + i\gamma$. The quantization for a few illustrative E_r are also displayed in Fig. 3(b).

To make our discussions clearer, we have added a short description in the main text to clarify this.

2. In the Method section, the assumption of calculating G_{1N} using the integral form in Eq. 26 is that β/N is vanishingly small. Yet in the main text, with $N = 100$, the β parameter is varied from 0 to 90, which is beyond this assumption. Does the result of G_{1N} calculated numerically by Eq. 25 deviate from Eq. 26 across the wide range of β ?

We thank the reviewer for raising this detailed but important question. In the Method section, we have considered two different parameter regimes with vanishing β/N (for $\beta \ll \beta_c$) and β proportional to N (for $\beta > \beta_c$) respectively. In this two regimes, different routes need to be considered to rewrite the summation of Eqs. (30,31) into integrals, which lead to completely different results valid in their respective regimes. Therefore an analytic solution for the intermediate regime with $\beta \sim \beta_c$ is hard to obtain, and we have approached this regime through numerical simulations. The result of G_{1N} is calculated numerically by directly solving the equation $G(E_r - H) = I$ with “ I ” being the identity matrix, and Eqs. (25,26) are to understand the physics behind the quantification in some simple limits.

As numerically shown in all our figures, the quantization of the response for all cases agree exactly with our theoretical predictions.

3. On page 3, the authors claim a quantized classical response distinct from existing topological Hall response exclusively in quantum systems. However, this statement completely ignores the abundant examples of various Hall responses realized in photonic and sonic systems which are also classical.

This is an important comment which is relevant to understanding the novelty of our work. Indeed, analogs of topological insulators have been realized in various photonic and sonic setups. Our

reviewer is certainly right in that there are classical topological modes in such classical systems, and they are already demonstrated in abundantly many experiments.

However, in classical systems, each topologically mode (which is protected by a band topological invariant) is free to be excited arbitrarily strongly. In other words, a robust edge mode can be excited weakly or strongly depending on the energy put into it. As such, although the number of classical topological modes is quantized, the physical “Hall response” is not, since the Kubo formula only applies to quantum systems with well-defined occupied eigenstates. Specifically, to the best of our knowledge, in such experiments on classical topological systems, the topological properties are characterized through measuring the topological boundary modes at certain frequencies, which are analogs to the single-particle topological boundary states at certain eigenenergies in an electronic material. As a comparison, the quantized Hall response is defined for the occupied states, and does not apply to photonic and sonic lattices, which are essentially classical.

In this work, one main novelty is that we discover a new approach and definition of a classically measurable response that is truly topologically quantized. The idea is that the spectral winding topology – which is distinct from band topology – does not require well-defined occupied energy eigenvalues. In fact, this freedom from the requirement of occupied states also makes it much more general, being applicable to classical systems like photonic, circuit and sonic systems, in addition to quantum lattices

In the revised manuscript, we have made this point more concrete by acknowledging several experiments in sonic and photonic systems and their associated topological signatures, and contrasting them with the new classical quantized quantity proposed in our work.

4. The inverse temperature and the scaling factor of boundary couplings are both denoted by β , which causes confusion.

We are sorry for having used confusing notations, and have now expressed the inverse temperature explicitly as $1/k_B T$ in the related discussion (which has been moved to the Supplemental Materials as the first reviewer suggests).

5. The explanation regarding winding number of -1 for the topological trivial case on page 5 seems not provided in the Supplemental Material.

We apologize for this potentially confusing point. One subtle caveat in our approach is that negative quantized values from our approach actually indicates the absence of amplification, and hence trivial winding topology. We are grateful that the referee pointed this out, and our revised manuscript has now clarified this subtlety in detail. This caveat in no way changes the conclusion that positive topological windings corresponds to positive quantized responses.

Specifically, following our referee’s suggestion, we have now added a short explanation of the negative value in the topological trivial case in the Method section. That is, under PBC there shall be a finite signal amplifying/decaying rate between sites N and 1 , as they are two neighbor sites in a transitional-invariant system. Yet our calculation shows that the amplifying/decaying rate shall approach zero when the system is tuned away from PBC, hence the derivative of its variation shall

be negative. More checking shows that there is no guarantee that this derivative must be -1. In any case, this is not really important overall, because we only established that signal amplification, if it does occur, is linked to a quantized spectral winding number. We also explained in the main text that whatever negative value one has, it only indicates the absence of the amplification and hence a trivial winding topology.

REVIEWERS' COMMENTS

Reviewer #1 (Remarks to the Author):

Report:

I have read the revision of "Quantized classical response from spectral winding topology" by Linhu Li and collaborators and their response. The authors have rearranged and improved the paper substantially. Overall, I think that all my questions and concerns have been solved perfectly.

Specifically, the authors give more strict proof about the Eq.(7) for nearest-coupling cases. And they explain the sub-chain physical picture more clearly, which convinces me that the authors have established an exact correspondence between spectral winding and quantized response defined in Eq.(8) for a general single-band model. To more complete, the authors have also examined the robustness of the quantized response against the disorder of various strengths and extended the conclusion derived from single-band case to more bands by numerical calculations in the Supplementary Materials.

Based on this, I recommend this nice paper for publication in Nature Communication.

Reviewer #2 (Remarks to the Author):

In the revised manuscript, the authors have addressed my comments adequately. The scope of the manuscript is also expanded, and I am convinced of the novelty of the revised manuscript. Therefore, I support that the manuscript accepted in Nature Communications. A recent work closely related to the current manuscript has been published in PRL 126, 216407 (2021), and I suggest that the authors acknowledge this work.

Reviewer #1 (Remarks to the Author):

Report:

I have read the revision of “Quantized classical response from spectral winding topology” by Linhu Li and collaborators and their response. The authors have rearranged and improved the paper substantially. Overall, I think that all my questions and concerns have been solved perfectly.

Specifically, the authors give more strict proof about the Eq.(7) for nearest-coupling cases. And they explain the sub-chain physical picture more clearly, which convinces me that the authors have established an exact correspondence between spectral winding and quantized response defined in Eq.(8) for a general single-band model. To more complete, the authors have also examined the robustness of the quantized response against the disorder of various strengths and extended the conclusion derived from single-band case to more bands by numerical calculations in the Supplementary Materials.

Based on this, I recommend this nice paper for publication in Nature Communication.

We are glad that referee 1 is now completely satisfied with our work.

Reviewer #2 (Remarks to the Author):

In the revised manuscript, the authors have addressed my comments adequately. The scope of the manuscript is also expanded, and I am convinced of the novelty of the revised manuscript. Therefore, I support that the manuscript accepted in Nature Communications. A recent work closely related to the current manuscript has been published in PRL 126, 216407 (2021), and I suggest that the authors acknowledge this work.

We are glad that referee 2 is also completely satisfied with our work, and have cited their published paper accordingly.